# Sex Differences in the Neuropsychiatric Effects and Pharmacokinetics of Cannabidiol: A Scoping Review

**DOI:** 10.3390/biom12101462

**Published:** 2022-10-12

**Authors:** Justin Matheson, Zoe Bourgault, Bernard Le Foll

**Affiliations:** 1Translational Addiction Research Laboratory, Centre for Addiction and Mental Health, University of Toronto, 33 Ursula Franklin Street, Toronto, ON M5S 2S1, Canada; 2Department of Pharmacology and Toxicology, University of Toronto, Toronto, ON M5S 1A8, Canada; 3Campbell Family Mental Health Research Institute, Centre for Addiction and Mental Health, Toronto, ON M5S 1A8, Canada; 4Dalla Lana School of Public Health, University of Toronto, Toronto, ON M5S 1A8, Canada; 5Department of Psychiatry, University of Toronto, Toronto, ON M5T 1R8, Canada; 6Institute of Medical Science, University of Toronto, Toronto, ON M5S 1A8, Canada; 7Acute Care Program, Centre for Addiction and Mental Health, Toronto, ON M6J 1H4, Canada; 8Department of Family and Community Medicine, University of Toronto, Toronto, ON M5G 1V7, Canada; 9Waypoint Research Institute, Waypoint Centre for Mental Health Care, Penetanguishene, ON L9M 1G3, Canada

**Keywords:** cannabidiol, sex differences, psychiatry, addiction, pharmacokinetics

## Abstract

Cannabidiol (CBD) is a non-intoxicating cannabinoid compound with diverse molecular targets and potential therapeutic effects, including effects relevant to the treatment of psychiatric disorders. In this scoping review, we sought to determine the extent to which sex and gender have been considered as potential moderators of the neuropsychiatric effects and pharmacokinetics of CBD. In this case, 300 articles were screened, retrieved from searches in PubMed/Medline, Scopus, Google Scholar, PsycInfo and CINAHL, though only 12 met our eligibility criteria: eight studies in preclinical models and four studies in humans. Among the preclinical studies, three suggested that sex may influence long-term effects of gestational or adolescent exposure to CBD; two found no impact of sex on CBD modulation of addiction-relevant effects of Δ⁹-tetrahydrocannabinol (THC); two found antidepressant-like effects of CBD in males only; and one found greater plasma and liver CBD concentrations in females compared to males. Among the human studies, two found no sex difference in CBD pharmacokinetics in patient samples, one found greater plasma CBD concentrations in healthy females compared to males, and one found no evidence of sex differences in the effects of CBD on responses to trauma recall in patients with post-traumatic stress disorder (PTSD). No studies were identified that considered the role of gender in CBD treatment effects. We discuss potential implications and current limitations of the existing literature.

## 1. Introduction

Cannabidiol (CBD) is a cannabinoid compound derived from *Cannabis sativa*, an annual herbaceous plant that grows in a variety of habitats and has a rich history of use for industrial, spiritual, medicinal, and recreational purposes [1,2,3]. CBD was first isolated from the cannabis plant nearly a century ago and was found to be devoid of most of the psychotropic effects of the presumed primary psychoactive component of cannabis, later identified as (–)-trans-Δ⁹-tetrahydrocannabinol (THC) [4]. In 1963, Mechoulam and Shvo discovered the structure and stereochemistry of CBD [5], which was instrumental in elucidating the final structure of THC in the following years [6,7]. Two decades later, the endocannabinoid system (ECS) was discovered, first by functional characterization [8] and then cloning of the human cannabinoid type-1 receptor (CB1R) [9]. We now know that the ECS is comprised of at least three important components: (1) two canonical receptors, CB1R and the cannabinoid type-2 receptor (CB2R), both seven-transmembrane domain G-protein coupled receptors (GPCRs); (2) at least two endogenous lipid transmitters (the endocannabinoids), both phospholipid derivatives containing a poly-unsaturated fatty acid moiety (arachidonate) and a polar head group, either ethanolamine in the case of anandamide (AEA) or glycerol in the case of 2-arachidonoylglycerol (2-AG); and (3) enzymes responsible for the synthesis and degradation of the endocannabinoids [10]. The most important and well-studied ECS enzymes include: fatty acid amide hydrolase (FAAH), which is responsible for the majority of AEA metabolism; monoacyl glycerol lipase (MAGL), responsible for the majority of 2-AG metabolism; NAPE-specific phospholipase D (NAPE-PLD), an important enzyme for synthesis of AEA; and diacylglycerol lipase (DAGL), an important enzyme for synthesis of 2-AG [10,11]. 

THC was shown to exert the characteristic “tetrad” effects (hypolocomotion, hypothermia, catalepsy, and antinociception) observed with other cannabinoid agonists via its actions at CB1R, though with lower maximal effect sizes than full receptor agonists, indicating that THC is a partial agonist at CB1R [12]. However, CBD lacked the tetrad effects of other cannabinoid agonists and had very little activity at CB1R; thus, its actions appeared to be less dependent on the ECS [13]. The lack of intoxicating and psychotomimetic effects of CBD, along with its long list of potential therapeutic effects (including anxiolytic, analgesic, anti-inflammatory, antipsychotic, anticonvulsant, antiemetic, and antineoplastic effects), have positioned CBD as an attractive candidate molecule for human trials across a range of potential indications, including neuropsychiatric disorders [13,14,15]. 

In the United States, the CBD product Epidiolex has been approved by the Food and Drug Administration (FDA) for the treatment of seizures associated with Lennox-Gastaut syndrome, Dravet syndrome, or tuberous sclerosis complex in patients 1 year of age or older, which is currently the only approved use of CBD [16]. The anti-epileptic effects of CBD gained public attention from anecdotal reports, especially the case of an American 5-year-old with Dravet syndrome named Charlotte, who was experiencing nearly 50 daily generalized tonic-clonic seizures before three months of treatment with a high-CBD strain of cannabis (named “Charlotte’s Web”) led to a 90% reduction in seizure frequency [17]. Over the past 5 years, a handful of pivotal randomized controlled trials were conducted, with nearly all demonstrating a significant reduction in seizure frequency in the CBD-treated groups compared to placebo [16]. The success of CBD as an anti-epileptic drug has fueled the search for additional indications for CBD, given its widespread effects and complex array of molecular targets. 

Studies in animal models of neuropsychiatric disorders (including both mental health and substance use disorders) have identified neuroprotective, anxiolytic, antidepressant, anti-reward, and antipsychotic-like effects of CBD mediated through diverse molecular targets [18,19,20]. Even though the ECS is a less important target for CBD than for THC, CBD does have some activity at CB1R, where it likely acts as a negative allosteric modulator [21], as well as at CBR2, where it acts as a partial agonist [22]. In addition, CBD appears to inhibit FAAH, leading to increased AEA levels [23]. Preclinical studies have suggested that CB1R and FAAH are important targets mediating the potential anxiolytic, antidepressant, and antipsychotic effects of CBD [18,20]. Another important molecular target of CBD is the serotonergic system, especially 5-hydroxytryptamine receptor 1A (5HT1A), which is thought to play an important role in mediating the potential anxiolytic and antidepressant-like effects of CBD [18,20]. The peroxisome proliferator-activated receptor gamma (PPAR-γ) is yet another important target, which is likely involved in both anxiolytic and anti-reward effects of CBD [18]. CBD has been shown to promote adult neurogenesis in animal models, e.g., [24,25], possibly by inhibiting FAAH and thus increasing AEA concentrations or by targeting PPAR-γ, which is a promising putative mechanism to explain its broad neuroprotective effects [19]. CBD has numerous other potential mechanisms of action, which includes GABAergic, dopaminergic, cholinergic, and glycinergic receptors; transient receptor potential cation channels (TRP channels, e.g., TRPV1, TRPA1); as well as orphan GPCRs such as GPR3, GPR6, and GPR55 [18,20] (see Figure 1 for schematic overview). 

A large body of converging evidence has explored CBD as a putative pharmacological treatment for most common mental health and substance use disorders, which has been reviewed extensively in numerous comprehensive narrative [14,20,27,28,29,30,31,32] and systematic or scoping reviews [33,34,35,36]. One recent systematic review and one recent scoping review of human studies have highlighted the limited clinical evidence. Khan et al. (2020) included case reports, case series, open-label trials, and both non-randomized and randomized controlled trials (RCTs), concluding based on 23 studies that there was moderate evidence for CBD or nabiximols (a 1:1 oromucosal formulation of THC:CBD) in the treatment of psychosis, cannabis use disorder (CUD), social anxiety disorder (SAD), autism spectrum disorder (ASD), and attention deficit hyperactivity disorder (ADHD), with weak evidence for insomnia, bipolar disorder, post-traumatic stress disorder (PTSD), and Tourette syndrome [33]. Kirkland et al. (2022) identified 16 RCTs or within-subjects human trials investigating CBD alone as a treatment option, and concluded that while there is not strong enough quality evidence to suggest use of CBD for any psychiatric disorder, the strongest evidence exists for treatment of psychotic symptoms (based on six included studies) and anxiety (based on three included studies) [36]. The authors also identified 54 trials registered at the time (as of 20 April 2021), indicating substantial interest in further trialing CBD across a range of psychiatric indications. 

There are multiple reasons for the continued interest in the therapeutic potential of CBD in neuropsychiatric disorders. First, CBD is generally very well tolerated. Limited adverse effects (AEs) were reported in healthy volunteers at single doses up to 6000 mg CBD or repeated dosing of 1500 mg twice daily, with most common AEs including diarrhea and headache [37]. Similarly, a systematic review of human trials found that CBD was well tolerated in patient populations, e.g., 4 to 6 weeks of treatment with 600 to 1000 mg CBD per day was well tolerated in patients with schizophrenia, while single doses of up to 900 mg CBD were well tolerated in patients with anxiety disorders [38]. Tolerability is a very important consideration for psychiatric drug development, as experiencing AEs is an important reason for non-adherence, e.g., non-adherence to antipsychotics in patients with psychosis-spectrum disorders [30]. Second, while THC has significant abuse liability that can lead to development of a CUD with prolonged exposure, CBD appears to be almost entirely devoid of abuse liability, although this may depend to some extent on route of administration as inhaled CBD may be more likely to produce subjective effects in human studies [36]. Given the high comorbidity of mental health and substance use disorders, psychiatric drugs lacking abuse liability are important to reduce the risk of misuse [30,34]. Finally, given the numerous molecular targets of CBD, it may be particularly effective at treating comorbid conditions. Typically, it is a hindrance when bioactive molecules lack selectivity or specificity for one molecular target, as this leads to off-target effects; however, CBD appears to be an example of a “polypharmacological agent” that is capable of modulating multiple targets involved in common signaling pathways to exert therapeutic effects across a range of symptoms and disease models [18]. For example, CBD has shown promising in reducing symptoms of psychosis-spectrum disorders and CUD, which are frequently comorbid, and thus it may be particularly useful in the treatment of this comorbidity (possibly via targeting CB1R, FAAH, and/or 5HT1A) [34].

While CBD remains a promising novel pharmacological strategy for treatment neuropsychiatric disorders, one important consideration that not yet received enough attention is the role of the biological construct of sex (and possibly the sociocultural construct of gender) in moderating effects of CBD. A significant body of literature, including evidence from both human and animal studies, has identified sex as an important modulator of ECS activity and moderator of analgesic [39], reward, and “tetrad” effects of cannabinoids (mainly THC and synthetic CB1R agonists such as WIN 55,212-2) [40]. This is thought to be primarily due to interactions between the ECS and gonadal hormones such as estrogens, with findings that gonadal hormones can modulate CB1R expression and downstream signaling, inhibit FAAH, and modulate circulating levels of endogenous cannabinoid transmitters [39,41]. Further, human studies have identified sex/gender differences in patterns and motivations of cannabis use, including use for therapeutic purposes [40]. Of relevance to the present review, recent cross-sectional studies have found that women are more likely to use CBD-only or CBD-dominant cannabis products across a range of patient and non-patient samples [42,43,44]. Another recent cross-sectional study of patients with chronic non-cancer pain found that women were significantly more likely to experience adverse effects of medical cannabis (specifically, central nervous system, gastrointestinal, musculoskeletal, and psychological AEs), while also consuming significantly higher monthly doses of CBD than men [45]. 

Since we were unable to locate any existing reviews of sex or gender considerations in CBD research, our goal was to conduct a scoping review of all available preclinical (non-human animal) and clinical (human) evidence of sex or gender influences on neuropsychiatric effects of CBD. Given that we have previously found a significant impact of participant sex on blood concentrations of THC after smoking cannabis in a human laboratory paradigm [46] and that multiple studies in rodents models have demonstrated sex differences in THC pharmacokinetics [40], we decided to include CBD pharmacokinetics as a potential endpoint of interest. We based our search strategy on recent systematic or scoping reviews mentioned earlier in the text [35,36], with the added dimension of sex or gender considerations. 

## 2. Methods

A search was conducted in PubMed/Medline, Scopus, and Google Scholar on 8 July 2022, with additional searches conducted in PsycInfo and CINAHL on 14 July 2022. There were no publication date restrictions on included articles, so any articles available in each searched database were eligible up until the noted search dates. Searches included the following terms using Boolean operators: (sex or gender) and (cannabidiol or CBD) and (psychiatry, psychiatric, anxiety, “post-traumatic stress disorder”, “panic disorder”, “generalized anxiety disorder”, “bipolar disorder”, mania, “mood disorder”, depression, “major depressive disorder”, “obsessive-compulsive disorder”, psychosis, schizophrenia, “psychotic disorder”, “substance use disorder”, addiction, alcohol, “alcohol use disorder”, cannabis, “cannabis use disorder”, tobacco, nicotine, “tobacco use disorder”, stimulant, “stimulant use disorder”, cocaine, “cocaine use disorder”, amphetamine, methamphetamine, opioid, “opioid use disorder”, or pharmacokinetics). The search terms used were adapted based on recent scoping or systematic reviews [35,36]. A modified search query was used in Google Scholar due to the character limit.

Included articles had to meet the following criteria: The article described results of a human study or in vivo animal study. Studies were excluded if they included only in vitro data.The study involved acute or repeated exposure to CBD alone. Studies that did not include exposure to CBD or to CBD only in combination with THC were excluded. An exception was made if the goal of co-administering CBD with THC was to determine the impact of CBD on addiction-related effects of THC, which provides evidence of the potential utility of CBD to treat CUD.The study included an endpoint related to mood or anxiety, psychosis, PTSD, OCD, addiction, or pharmacokinetics of CBD. Articles were excluded if they contained no relevant endpoints (e.g., included molecular or cellular endpoints without clear relevance to psychiatric disorders; included endpoints related only to pain; or measured only cannabinoid “tetrad” effects).The article included a sex- or gender-based analysis (no studies were identified that assessed the impact of the sociocultural construct of gender on CBD outcomes, so we focus only on sex moving forward). A study was considered to have a sex-based analysis when one of two conditions was met: (1) sex was analyzed as a main effect when the outcome of interest was CBD concentrations or pharmacokinetics (e.g., a *t*-test to determine if males and females differed in physiological concentrations of CBD); or (2) the study had a factorial design with sex as a factor that entered into an interaction term with a drug exposure or treatment factor (e.g., a two-way ANOVA to determine if there was an interaction between sex and CBD dose group on a relevant endpoint). This definition was based on previous literature describing appropriate reporting of sex differences [47,48].The article was a primary, peer-reviewed research article available in English. Review articles and other non-primary articles were excluded, as well as those not available in English.

Using Covidence systematic review software (Veritas Health Innovation, Melbourne, Australia; available at www.covidence.org (accessed on 7 July 2022)), two reviewers (JM and ZB) first screened articles by title/abstract and then reviewed full-text articles for inclusion based on the described eligibility criteria. When conflicts arose between reviewer ratings of individual articles, the two reviewers were able to reach consensus without consulting a third party. The search terms, databases of interest, and article eligibility criteria were decided on prior to initiating the search process in an unstructured review protocol. The authors created a Word template for extraction of data which included reference (author, year), species (or study population, for human studies), sex (*n*), CBD dosing, relevant endpoints (which were not pre-specified due to the breadth of literature covered), and results. This information was compiled into two tables (one for animal studies, one for human studies) and synthesized qualitatively in the main text that follows. 

## 3. Results

### 3.1. Screening of Included Articles

After removing duplicates, a total of 300 individual articles were screened, with 30 moving forward for full-text review (see Figure 2 for PRISMA flowchart). Of the 30 studies, 18 were excluded: 12 did not include a sex-based analysis (i.e., no between-sex comparison or interaction term that included sex and treatment), two included only in vivo data, two included no relevant endpoints, one included CBD exposure only in combination with THC, and one was not English language. This left 12 articles that met our eligibility criteria: eight studies in animal models and four studies in humans. Initially, our plan was to present only results from analyses that included sex as a “discovery” variable (i.e., as a main effect or part of an interaction term), in line with a recent review of inclusion of sex as a biological variable in neuroscience and psychiatric papers [47]. However, since there were already so few studies identified for this review, and since unfortunately most preclinical studies did not consistently include sex as a discovery variable or did not clearly present between-sex comparisons when there were significant sex-by-drug treatment condition interactions, we present here a mix of between-sex and within-sex comparisons. 

### 3.2. Preclinical (Non-Human Animal) Evidence

Eight animal studies were identified that involved exposure to CBD, inclusion of an endpoint relevant to neuropsychiatric disorders (including pharmacokinetics), and inclusion of sex as a discovery variable in the analysis. We had initially planned to group findings by endpoints, but since there were fewer included studies and more heterogeneity in endpoints than expected, we instead group by behavioural endpoints assessed either after gestational or developmental exposure to CBD or acute/chronic exposure to CBD. Studies assessing sex differences in CBD pharmacokinetics are presented separately. See overview of studies in Table 1.

#### 3.2.1. Long-Term Behavioural Endpoints after Gestational or Developmental Exposure to CBD

Two studies assessed behavioural effects of maternal gestational exposure to CBD in offspring [50,51] and one study assessed long-term behavioural effects of CBD exposure in the adolescent period [52]. 

Maciel et al. (2022) administered 3 mg/kg CBD subcutaneously to pregnant CD1 mice from gestational day (GD) 5 to post-natal day (PND) 10, and observed no statistically significant sex differences in behavioural task performance across a number of tasks (open field, sucrose preference, marble burying, nestlet shredding, forced swim) [50]. However, in this same study, gestational exposure to CBD from GD 5 to 18.5 led to increased brain concentrations of CBD in female embryos compared to male embryos. Wanner et al. (2021) administered 20 mg/kg CBD orally to pregnant Agouti viable yellow mice for nine weeks, starting at two weeks prior to mating, and found a significant sex by treatment interaction when examining spontaneous alternation percent in the Y-maze (a measure of spatial memory) [51]. While the authors did not conduct a between-sex comparison, this interaction appeared to be driven by a statistically significant effect of treatment in females only, where female CBD-exposed offspring had an increase in spontaneous alternation percent (indicating better performance) compared to vehicle-exposed female offspring, with no similar treatment effect in males. A similar trend was observed in the marble burying test, where CBD-exposed females buried more marbles (indicating increased anxiety and/or compulsive behaviour) than vehicle-exposed females, with no treatment effect in males, though the sex by treatment interaction was not significant in this case [51]. Finally, Kaplan et al. (2021) administered 20 mg/kg CBD via twice-daily intraperitoneal injections to C57BL/6 mice from PND 25 to 45 and assessed behavioural outcomes starting at PND 60 to examine long-term effects of adolescent exposure to CBD [52]. They found a significant sex by treatment interaction when examining number of errors in the Barnes maze (a spatial memory task), where CBD-treated females made fewer errors than vehicle-treated females and fewer errors than CBD-treated males at trend level (*p* = 0.06), with no effect of CBD or sex on locomotor activity in an open field test or anxiety-related behaviours in the elevated plus maze. However, they did find that CBD reduced weight gain in females (but not males) [52]. 

Taken together, these findings suggest that sex may have a small impact on long-term behavioural effects of prenatal or developmental exposure to CBD, but that this impact is likely dependent on the behavioural endpoints assessed. Interestingly, both gestational and adolescent exposure to CBD were associated with an increase in spatial memory performance in female, but not male, mice. 

#### 3.2.2. Behavioural Endpoints after Acute or Repeated Exposure to CBD 

Four studies assessed the behavioural effects of acute or repeated exposure to CBD in adolescent or adult rodents. Two of these studies focused on the potential of CBD to modulate addiction-related effects of THC [53,54] and the other two focused on antidepressant-like effects of CBD [55,56]. 

**Table 1 biomolecules-12-01462-t001:** Overview of included articles detailing studies in animal models.

Reference	Species Sex (*n*)	CBD Dosing	Relevant Endpoints	Results
Child and Tallon, 2022 [57] ^1^	Sprague-Dawley rats Female *n* = 6 per dose group Male *n* = 6 per dose group	Daily oral gavage of 0, 30, 115, or 230 mg/kg CBD for 28 days	PK parameters at 115 mg/kg/day dose only (Day 1 and Day 28): (1) T_max_ (2) C_max_ (3) AUC_0–24h_ (4) K_a_ (5) K_e_ Tissue CBD concentrations (at all three active doses): (6) Adipose (7) Muscle (8) Liver	Greater Day 28 AUC in females (3) Higher liver CBD concentrations (8) in females at 115 mg/kg/day dose (n.s. at other doses) No sex differences in other PK parameters (1,2,4,5) or adipose or muscle tissue concentrations (6,7)
Hempel et al., 2018 [53] ^2^	Wistar rats Female *n* = 8–9 per dose group Male *n* = 8–9 per dose group	0.075 or 0.75 mg/kg i.p. CBD, alone or in combination with 0.75 mg/kg THC	Conditioned taste avoidance and place aversion	No sex difference in either procedure
Kaplan et al., 2021 [52]	C57BL/6J mice Female *n* = 5 per group Male *n* = 8 per group	Twice-daily i.p. injections of 20 mg/kg CBD or vehicle from PND 25 to 45	(1) Locomotor activity (open field), PND 60 (2) Anxiety-related behaviours (elevated plus maze), PND 65 (3) Spatial memory (Barnes Maze), PND 70–74	Significant sex by condition interaction for number of errors in the Barnes Maze (3): CBD-treated females made fewer errors than vehicle-treated females, and fewer errors than CBD-treated males (trend at *p* = 0.06) No significant impact of sex on locomotor activity or anxiety (1,2)
Ledesma-Corvi et al., 2022 [55] ^3^	Sprague-Dawley rats Female *n* = 8–9 per group Male *n* = 8–9 per group	Daily i.p. injection of 10 mg/kg CBD (or vehicle, fluoxetine, or ketamine) for 7 days, during adolescence and then re-exposure for another 7 days during adulthood	Antidepressant-like responses assessed during (acute effects) and immediately following (repeated effects) adolescent and adult CBD exposure: (1) Forced-swim test (FST) (2) Novelty-suppressed feeding (NSF) test (3) Sucrose preference test	Significant sex by treatment or sex by treatment by early-life condition interactions: (1) Significant effect of CBD in males, but not females, for climbing (acute effect, during adolescence) and immobility (repeated effect, during adulthood) during the FST No other evidence of sex differences in CBD effects (2,3)
Maciel et al., 2022 [50] ^4^	CD1 mice Female *n* = 10 or 26 in CBD group Male *n* = 7 or 17 in CBD group	Gestational exposure to 3 mg/kg s.c. CBD from GD 5 to 18.5 (embryonic brain CBD) or GD 5 to PND 10 (behavioural tests)	(1) Embryonic brain CBD concentrations; Behavioural tests of offspring of dams gestationally exposed to CBD (PND 80–120): (2) Open field (3) Sucrose preference test (4) Marble burying (5) Nestlet shredding (6) Forced swim test	Embryonic brain concentrations of CBD (1) were significantly higher in females No statistically significant between-sex effects or sex by condition interactions in behavioural tests (2–6)
Silote et al., 2021 [56]	Experiment 1 (EXP1): Swiss mice Female *n* = 5–10 per group Male *n* = 4–8 per group EXP2: C57BL/6 mice Female *n* = 9–13 per group Male *n* = 7–8 per group EXP3: Flinders Sensitive Line (FSL) and Flinders Resistant Line (FRL) rats Female *n* = 4–10 per group Male *n* = 6–12 per group	EXP1/2: I.p. injection of vehicle or CBD (3, 10, or 30 mg/kg) EXP3: I.p. injection of vehicle or CBD (10, 30, 60 mg/kg)	(1) Elevated plus maze (EPM) (2) Tail suspension test (TST) (3) Open field test (OFT)	EXP1: Significant sex by treatment interaction in TST (2), driven by decreased immobility time in CBD-treated Swiss male mice (all doses), with no effect in females; no sex by treatment interaction in the EPM (1) EXP2: No significant effect of sex in C57BL/6 mice (1,2) EXP3: Significant sex by treatment by time interaction for immobility in FST, possibly due to difference in direction of CBD 30 mg/kg effect (reduced immobility time in FSL males, increased immobility time in FSL females, though neither effect was significant) Significant sex by treatment (and by time) interaction for total distance travelled in OFT
Wakeford et al., 2017 [54] ^5^	Long-Evans rats Female *n* = 2 Male *n* = 3	I.v. infusion of 1:1 or 1:10 ratio of CBD to THC (dose dependent on total amount of THC self-administered)	IVSA of THC	No significant effect of sex reported
Wanner et al., 2021 [51]	Agouti viable yellow mice Female *n* = ? Male *n* = ?	Sexually mature nulliparous female mice exposed to 20 mg/kg oral CBD or vehicle for nine weeks, starting two weeks prior to mating	Behavioural procedures in F1 offspring, starting at 12 weeks of age: (1) Y-maze spontaneous alternation (2) Marble burying	Significant sex by treatment interaction for the Y-maze test (2), which seemed to be driven by increased % spontaneous alternations in CBD-exposed females compared to vehicle-exposed females, while there was no effect in males No significant sex by treatment interaction for marble burying ^6^

AUC, area under the curve; C_max_, maximum concentration; GD, gestational day; i.p., intraperitoneal; i.v., intravenous; IVSA, intravenous self-administration; K_a_, absorption rate constant; K_e_, elimination rate constant; PK, pharmacokinetic; PND, post-natal day; s.c., sub-cutaneous; T_max_, time to maximum concentration. ^1^ Both within-sex and between-sex comparisons were conducted in this study; only between-sex comparisons are presented here. ^2^ Sex difference analysis is described in the methods (sex entered as a between-subjects factor in three-way ANOVAs) but not presented in the results, except for a statement that there were no sex differences for any comparison in the place or taste conditioning. ^3^ Outcome data were analyzed using a three-way ANOVA with the factors sex (male or female), drug treatment (vehicle, fluoxetine, ketamine, or CBD), and early-life conditions (maternal deprivation or controls); no between-sex comparisons were clearly described in the results, so significant two- or three-way interactions involving the factor sex are presented here, but only if there was a significant effect of CBD (vs. vehicle) in at least one sex. ^4^ While there are within-sex differences described in this study, only results that clearly stated a statistical effect of sex (main effect of sex for embryonic brain CBD concentrations; interaction of sex with gestational exposure group for behavioural outcomes) are presented here. ^5^ While there was a two-way ANOVA (sex by CBD:THC ratio) described in the methods, the result was not presented, except for a statement that there were no sex differences for any comparison (therefore, date were pooled across sex). ^6^ Note that there was a main effect of sex.

Both studies of CBD modulation of THC effects were conducted by the same group. In their first study, Wakeford et al. (2017) found no sex differences in the effect of CBD pre-treatment on intravenous self-administration of THC in Long-Evans rats (intravenous administration at a 1:1 or 1:10 CBD:THC ratio), though only 3 males and 2 females were included in this experiment, and there was no overall effect of CBD, regardless of sex [54]. In a subsequent study, Hempel et al. (2018) found no sex differences in CBD place aversion or taste avoidance conditioning (though CBD, administered intraperitoneally at 0.075 or 0.75 mg/kg, failed to produce conditioned place aversion or taste avoidance on its own, regardless of sex) or in CBD’s effect on THC place aversion or taste avoidance conditioning [53]. In both studies, there was no effect of CBD alone and the sex difference analysis was not shown, limiting conclusions that can be drawn. 

Ledesma-Corvi et al. (2022) administered daily 10 mg/kg CBD intraperitoneal injections for seven days during adolescence in Sprague-Dawley rats, then re-exposed the rats for another seven days in adulthood, and tested antidepressant-like effects during both treatment phases on day 1 of CBD treatment (acute effects) and immediately after treatment (repeated effects) [55]. Approximately half of the rats were exposed to maternal deprivation, and there were also cohorts who received vehicle, fluoxetine, or ketamine, which led to complex three-way interactions of sex, treatment group, and early-life condition. This, coupled with the fact that the authors did not report between-sex comparisons when there were significant interaction terms involving sex, made it challenging to interpret any potential sex differences in the antidepressant effects of CBD. During adolescence, there were significant acute and repeated effects of CBD on immobility time (decreased relative to vehicle) and climbing (increased) in the forced swim test, in male control (but not maternally deprived) rats, which was not true in female rats. Similar effects on immobility time and climbing were observed in male adults, though only after repeated exposure to CBD (no acute effect). CBD did not have effects on the novelty-suppressed feeding or sucrose preference tests in either adolescence or adulthood, regardless of early-life condition, with the exception of a small decrease in distance travelled in the novelty-suppressed feeding test in male maternally-deprived rats [55]. Silote et al. (2021) administered a single intraperitoneal injection of 3, 10, or 30 mg/kg CBD to both Swiss and C57BL/6 mice and Flinders Sensitive Line (FSL) and Flinders Resistant Line (FRL) rats (who were additionally administered a 60 mg/kg dose), and then tested antidepressant-like effects [56]. Of note, once again, the authors did not present between-sex comparisons, so findings of sex by treatment interactions are described. In Swiss mice, there was a significant sex by treatment interaction for immobility time in the tail suspension test, where male Swiss mice demonstrated less immobility time at all CBD doses tested related to vehicle, with no effect in female Swiss mice or in C57BL/6 mice of either sex. There was no sex by treatment interaction for endpoints related to elevated plus maze performance. In FSL and FRL rats, there was a significant sex by treatment by time (1 h or 2 h prior to testing) interaction for immobility in the forced swim test and both a sex by treatment by time and two-way sex by treatment interaction for total distance travelled in the open field test. The relevance of these interaction terms were not clear, as there was few statistically significant within-sex comparisons, though it appeared that CBD had a bimodal effect on immobility time in the forced swim test in FSL females, where CBD increased immobility time when administered 1 h prior to testing, and decreased immobility time when administered 2 h prior to testing [56]. 

Taken together, these findings suggest that there may be no sex differences in the influence of CBD on addiction-relevant effects of THC, though this conclusion is tempered by the lack of overall effects of CBD regardless of sex. In addition, CBD appears to have antidepressant-like effects in male rodents, but not female rodents, though this difference is likely dependent on numerous factors including age, dosing paradigm, and species or strain of rodent. However, this conclusion is limited by the lack of between-sex comparisons presented in the two studies included.

#### 3.2.3. Pharmacokinetics of CBD

One study examined sex differences in CBD pharmacokinetics after acute and repeated dosing in adult rats [57]. Child and Tallon (2022) administered 30, 115, or 230 mg/kg CBD daily by oral gavage for 28 days and assessed CBD pharmacokinetics on day 1 and day 28 [57]. At the 115 mg/kg dose, there was a significantly higher day 28 AUC_0–24h_ in females (50,561 ± 15,090) compared to males (28,602 ± 8486). There were also two notable within-sex difference: the C_max_ of CBD increased from day 1 to day 28 in females (1648 ± 330 to 2242 ± 529 ng/mL), while the t_max_ of CBD decreased from day 1 to day 28 in males (8:25 ± 0:32 to 7:47 ± 0:16 hr:min). There was no sex difference in the absorption rate constant or the elimination rate constant. The authors also explored tissue-specific concentrations of CBD. Overall, there was a significant sex by dose interaction for liver tissue concentrations of CBD, but no effect for adipose or muscle tissue concentrations. Specifically, at the 115 mg/kg dose, liver concentrations of CBD were higher in females (1.59 ± 0.31 mg/kg) compared to males (0.30 ± 0.04 mg/kg). Though there was no between-sex comparison described, the authors subsequent reported that both the absorption rate constant and the elimination rate constant were negative correlated with adipose tissue CBD concentrations in males and that the decrease in CBD t_max_ from day 1 to day 28 was positively correlated with both muscle and liver CBD concentrations in males, whereas, no correlations were significant in females. Finally, as described earlier in the text, Maciel et al. (2022) reported significantly greater embryonic brain concentrations of CBD in females (0.039 ± 0.0013 nmoles/g) compared to males (0.0096 ± 0.0014 nmoles/g) [50]. 

Taken together, these data suggest that CBD concentrations are higher in females compared to males, with some tissue-specific pharmacokinetic relationships. 

### 3.3. Clinical (Human) Evidence

See overview of studies in Table 2.

#### 3.3.1. Behavioural or Clinical Endpoints after Exposure to CBD

Only a single human study was identified that presented a sex-based analysis of CBD behavioural or clinical effects in humans. Bolsoni et al. (2022) recruited 25 females and 8 males diagnosed with PTSD to test the effect of a single oral dose of 300 mg CBD vs. placebo (parallel design) on subjective and physiological responses to recalling trauma [58]. The authors did not find evidence that sex had an impact on the effect of CBD, though it should be noted the sex difference analysis was not described in the methods or results but mentioned as an exploratory outcome in the discussion (the data were not shown in the manuscript) [58].

#### 3.3.2. Pharmacokinetics of CBD

Three studies in humans have examined sex differences in CBD pharmacokinetics [59,60,61]. Consroe et al. (1991) recruited 6 females and 8 males diagnosed with Huntington’s disease to receive 6 weeks of daily treatment with 10 mg/kg CBD (corresponding to an average of 700 mg daily total CBD) or placebo in a crossover design [59]. There was no significant sex difference in CBD plasma concentrations averaged over the 6 weeks of treatment (mean ± standard error: 8.9 ± 0.9 for males, 7.1 ± 1.3 ng/mL for females) or in CBD concentrations at the one-week washout period. Contin et al. (2021) recruited 24 females and 19 males diagnosed with pharmacoresistant epilepsy (Dravet syndrome or Lennox-Gastaut syndrome) and collected data over a variable period lasting between one and 12 months, with participants receiving between 4.6 and 22.8 mg/kg of CBD daily in an open trial design [60]. There was no significant difference in the ratio of morning trough CBD plasma concentration to weight-adjusted daily dose, though the authors noted that the female subgroup was significantly older than the male subgroup, and they did observe a significant effect of age on CBD ratio, which was higher in patients ≥18 years compared with those under 18 years. Finally, Knaub et al. (2019) recruited 8 female and 8 male healthy volunteers to receive a single oral dose of 25 mg CBD in one of two formulations, either a medium-chain triglycerides formulation (MCT-CBD) or a novel self-emulsifying drug delivery system formulation (SEDDS-CBD) [61]. For the MCT-CBD formulation, both the AUC_0–8h_ and the AUC_0–24h_ were significantly higher in females compared to males, whereas the t_max_ was slightly faster in males than females for the SEDDS-CBD formulation. While the C_max_ of CBD was more than twice as high in females compared to males with the MCT-CBD formulation, this did not reach statistical significance.

Taken together, there appears to be no significant sex differences in CBD concentrations in patient populations, while healthy female volunteers may reach greater concentrations of CBD compared to health male volunteer, depending on the formulation of CBD administered. 

## 4. Discussion

The goal of this scoping review was to determine if sex or gender impact the neuropsychiatric effects or pharmacokinetics of CBD. Broadly speaking, we found few studies that included any mention of sex-based analysis (and zero human studies including the sociocultural construct of gender as a relevant variable), with even fewer studies including sex as a discovery variable (i.e., testing for a sex difference or a drug treatment by sex interaction). Based on the eight animal studies and four human studies reviewed, we highlight three very tentative conclusions that are each based on data from a different pair of two studies using different experimental approaches: (1) extended gestational or adolescent exposures to CBD were associated with improved spatial memory performance when tested later in life in female, but not male, mice; (2) acute or repeated exposure to CBD in adolescence or adulthood seems to have an antidepressant-like effect in male, but not female, mice and rats; and (3) acute or repeated exposure to CBD appears to lead to greater CBD concentrations in healthy female, but not male, rats and humans, though no sex differences were detected in CBD concentrations in humans diagnosed with Huntington’s disease or pharmacoresistant epilepsy.

### 4.1. Sex Differences in the Behavioural Effects of Gestational or Developmental Exposure to CBD

The marketing of cannabis and related products as “safe” and “natural” has led to increased use during pregnancy, including the use of CBD products for putative anti-emetic, anxiolytic, and analgesic effects [62]. While human and animal evidence has found potential adverse effects of prenatal cannabis or THC exposure on a range of developmental outcomes (e.g., low birth weight, cognitive and neuropsychiatric abnormalities) [63], less is known about the effect of prenatal exposure to CBD. Two relevant studies were included in our review. First, Maciel et al. (2021) found that gestational exposure to both THC and CBD altered repetitive and hedonic behaviours in adult mice and prevented fluoxetine from decreasing immobility (an antidepressant-like effect), which was rescued by inhibition of FAAH [50]. While there was no statistical evidence of between-sex differences, there were within-sex differences; specifically, females perinatally exposed to CBD exhibited significantly more repetitive behaviours (marble burying), while males had significant increased sucrose preference [50]. Of note, sex significantly impacted embryonic brain concentrations of CBD following gestational exposure, which could in part contribute to sex differences in observed behaviours [50]. Second, Wanner et al. (2021) found that gestational exposure to CBD led to increased repetitive or anxiety-like behaviours (marble burying) and also improved spatial working memory, with effects generally more apparent in female offspring compared to males [51]. Thus, sex does seem to play a role in determining long-term effects of gestational exposure to CBD, possibly by altering uptake of CBD into the developing fetus. 

The only currently approved indication for CBD is pharmacoresistant epilepsies, which primarily impact children. As a result of the perception of safety in pediatric populations, there has been increasing interest from patients and clinicians for off-label use of CBD in children and adolescents [64], leading to concerns about long-term safety. Kaplan et al. (2021) tested whether prolonged adolescent exposure to CBD would impact behaviours in adulthood to address this gap in knowledge and found no significant negative impact of CBD on locomotor activity, anxiety-like behaviours, or spatial memory [52]. Interestingly, there was a sex-specific effect of adolescent exposure to CBD on spatial working memory when tested in adulthood, where female mice had greater performance [52], which is in line with the findings of Wanner et al. (2021) of increased spatial memory performance in adult female mice who were perinatally exposed to CBD [51]. More work is needed to determine if adolescent exposure to CBD has long-term adverse effects and whether sex plays a moderating role, especially in humans. In addition, the female-specific positive effect of gestational or adolescent exposure to CBD on spatial working memory when tested in adulthood should be clarified. While this finding may be related to the sex difference observed in uptake of CBD into the fetal brain, it is also possible that this finding reflects an overall sex difference in the sensitivity of the developing brain to perturbations in neurocircuitry underlying spatial memory. For example, maternal exposure to sound stress and forced swimming has been found to cause spatial memory deficits in male offspring, with no effect seen in female offspring, an effect at least partly mediated by a serotonergic mechanism involving 5HT1A receptors [65].

### 4.2. Sex Differences in the Neuropsychiatric Effects of CBD

We found very few studies that examined whether sex moderates potential therapeutic effects of CBD in animal models of neuropsychiatric disorders or in human trials. Results obtained from included studies are discussed here for the three endpoints that were captured by our review: CBD modulation of THC effects (and by extension, potential of CBD for treatment of CUD and other substance use disorders), depression, and PTSD. Relevant data from excluded studies are considered here to provide additional context. 

First, two studies found no sex differences in the effects of CBD on modulation of IV self-administration of THC in rats [54] or THC place aversion or taste avoidance conditioning in rats [53], though both studies were limited by small sample sizes and a lack of CBD effects in the combined sex sample. CBD has shown promise in treating CUD in human trials, either on its own [66] or in combination with THC (i.e., nabiximols) [67,68,69,70,71]. As previous animal studies have found anti-craving effects of CBD in preclinical models of addiction [72], more work is needed to characterize sex differences in the potential role of CBD in reducing symptoms of CUD and other substance use disorders. For example, one study that was excluded from our review found that repeated exposure to CBD was able to reduce ethanol intake in male mice at lower doses (30 and 60 mg/kg) and reduced ethanol intake in both sexes only at a higher dose (90 mg/kg), suggesting some degree of sex difference in the potential of CBD to reduce binge drinking [73]. No published work was identified that considered whether sex impacted the effects of CBD on symptoms or models of tobacco, opioid, or stimulant use disorders, which are all potential CBD indications under investigation in ongoing or future human trials [36]. 

Second, two studies found evidence of antidepressant-like effects of CBD in male (but not female) rats and mice [55,56]. Of note, these studies found that the antidepressant-like effects of CBD were also influenced by species, strain, early life exposure to maternal deprivation, CBD exposure paradigm (acute vs. repeated), and age of the rodents [55,56], which speaks to the difficulty of assessing sex differences against a backdrop of multiple other potential moderating factors. These results are consistent with one previous study that found antidepressant-like effects of CBD across males of two different genetic models of depression [Wistar Kyoto (WKY) and Flinders Sensitive Line (FSL) rats], while CBD had antidepressant-like effects in females of only one genetic model (WKY rats) [74]. However, one single-sex study did find an antidepressant-like effect (reduced immobility time in the tail suspension test) of CBD in early-life stressed female mice [75]. As noted in a recent scoping review [36], there are no published human trials of CBD for the treatment of depression, though there are at least three currently registered trials. It will be prudent to consider sex as a relevant variable in these trials to determine if the male-specific antidepressant-like effects of CBD are replicated in humans. 

Finally, one study in humans diagnosed with PTSD found no evidence for sex moderating the effects of CBD on subjective and/or physiological responses to recalled trauma [58], though there was a significant sex imbalance in the sample (25 females, 8 males). Since there are known sex differences in the neurobiology of PTSD (including sex differences in endocannabinoid system modulation of stress responses, emotional memory, and fear extinction) [76], more work is needed to characterize sex differences in the potential role of CBD in treating PTSD. A recent preclinical study found that CBD reduced contextual memory and generalized fear memory, while enhancing extinction learning, in female mice [77]. However, this was a single-sex study, so it is unclear what relevance this finding has for understanding sex differences in PTSD-relevant effects of CBD. Currently, only a single case series has demonstrated potential clinical utility of CBD in reducing symptoms of PTSD [78], though there are multiple registered trials either planned or ongoing [36]. 

In their recent scoping review of potential clinical utility of CBD in treating psychiatric disorders, Kirkland et al. (2022) identified 16 relevant trials, with a majority of included trials either for psychotic disorders (*n* = 6) or anxiety disorders (*n* = 3) [36]. Interestingly, no studies were included in the present review that included either of these endpoints. However, a handful of studies that were excluded from our review have suggested that sex may influence the effects of CBD on neural mechanisms of psychosis in preclinical models. For example, using a maternal immune activation (poly I:C) model of schizophrenia, one study found that CBD normalized muscarinic signaling (muscarinic M1/M4 receptor binding density) in male, but not female, poly I:C offspring [79]. Another study found that CBD had sex-dependent effects on cortical neural activity in the rat methylazoxymethanol acetate (MAM) model of aberrant neural signaling in schizophrenia [80]. In terms of sex differences in potential anxiolytic effects of CBD, one study that was excluded from the present review reported no sex differences in acute effects of CBD on anxiety-like behaviours, though there was no anxiolytic effect of CBD alone in the combined sex sample, and there was no formal statistical test of sex differences [81]. One other single-sex study did find an anxiolytic effect of CBD in females which was comparable to other findings in males [82], though the lack of sex comparison limits conclusions that can be made.

Taken together, the existing evidence does suggest a role of sex in moderating the neuropsychiatric effects of CBD, though the evidence is too sparse to make any firm conclusions regarding which specific endpoints are likely impacted. The only sex difference that emerged in more than one study was the potential for antidepressant-like effects of CBD in male rodents, but not female rodents. More work will be needed to characterize potential sex influences on mechanisms of antidepressant-like effects of CBD in animal models and to see if this finding translates in human clinical trials. 

### 4.3. Sex Differences in CBD Pharmacokinetics

There are well known sex differences in THC pharmacokinetics, at least in animal models. In 1991, Narimatsu and colleagues found that female rats primarily metabolize THC to its psychoactive and equipotent metabolite 11-OH-THC, while THC is metabolized to a greater variety of largely non-psychoactive metabolites in males [83], which is likely at least partially responsible for sex differences in behavioural effects of THC [84]. Since then, several studies have replicated and expanded these findings, consistently finding greater THC and 11-OH-THC concentrations in female animals compared to male animals [40], with some evidence of tissue-specificity, e.g., greater magnitude sex differences in brain 11-OH-THC [85]. While the human evidence is more mixed, at least two studies have found a dose- and route-dependent sex difference, mirroring the greater THC concentrations in females observed in animal studies [86,87]. 

In the present review, we found some parallels to the THC literature, where CBD concentrations tended to be higher in females, though the evidence was mixed in humans. In mice, higher embryonic brain CBD concentrations were found in females compared to males after gestational exposure to CBD, which the authors suggested could be due to enzymatic differences driving sexual differentiation [50]. In adult rats, CBD concentrations were significantly higher in females compared to males (after 28 days of dosing, but not after the first day of dosing), and there was a tissue-specific sex difference in distribution, with greater liver CBD concentrations in females compared to males [57]. In a single-dose trial with healthy human volunteers, there was a significant sex difference in CBD concentrations with at least one formulation, with greater AUC in females compared to males [61]. However, two additional human studies in patients with Huntington’s disease [59] or pharmacoresistant epilepsy (Dravet syndrome or Lennox-Gastaut syndrome) [60] failed to find sex differences in CBD concentrations after repeated/long-term dosing. While the study was excluded because CBD was not administered alone, Nadulski et al. (2005) provided additional support for the finding of greater CBD concentrations in females; females had higher AUC of CBD than males after receiving a single oral dose of an extract of cannabis containing 10 mg THC and 5.4 mg CBD [86].

As findings of greater CBD concentrations associated with the female sex parallel the THC literature, it will be important to continue to monitor whether sex influences CBD pharmacokinetics in human trials. Increased plasma concentrations of CBD in female patients treated with CBD could drive an increase in adverse effects, which has already been found in a cross-sectional study [45].

### 4.4. Potential Mechanisms of Sex Differences in CBD Effects and Pharmacokinetics

Understanding the basis of sex differences in drug effects is complex and involves integrating different lines of inquiry. Typically, this is approached by working backwards: first identifying a sex difference in a behaviour or other endpoint, then determining whether there are sex differences in mechanisms mediating the endpoint, and then identifying how (or if) sexual biology influences this mechanism. Sex differences ultimately arise via actions of sex chromosomes, gonadal hormones, and/or other sex-biased contingent factors such as epigenetics. The reader is directed to an excellent review by Becker and Chartoff (2018) for a much more detailed overview of how sex differences arise in neural mechanisms mediating reward and addiction [88]. Due to the diverse effects and molecular targets of CBD, there is an extremely large number of possibilities for sex differences in CBD effects. We thus speculate on a few potential mechanisms below that are of particular relevance to the findings discussed in this review.

Though the ECS is a less important target for CBD than for THC, there are some proposed neuropsychiatric effects of CBD that are likely mediated through the ECS, which may be influenced by sex. For example, increasing AEA levels by inhibiting FAAH is a potential mechanism through which CBD promotes neurogenesis, which could partly underlie the observed anxiolytic- and antidepressant-like effects of CBD [19]. A relevant sex difference has been previously demonstrated with the FAAH inhibitor URB597, which was less able to reduce anxiety-like symptoms in female rats exposed to chronic unpredictable stress compared to male rats [89]. Human studies have found that both FAAH and AEA levels fluctuate across the menstrual cycle [90,91], which could be related to the discovery of an estrogen response element present in the mouse *Faah* gene [92]. Taken together, circulating levels of estrogens may directly impact FAAH expression and activity, which could in turn impact neuropsychiatric effects (e.g., anxiolytic- or antidepressant-like effects) of CBD mediated through FAAH. This could be one reason for the observed findings that CBD has more consistent antidepressant-like effects in male rodents compared to females. 

It is equally possible that there are sex differences in targets outside the ECS that lead to differences in neuropsychiatric effects of CBD. There are numerous important sex differences in neurocircuitry underlying animal models of fear, arousal, social avoidance, learned helplessness, and anhedonia, which collectively speak to sex differences in mechanisms of anxiety, depression, and trauma-related neuropsychiatric disorders [93]. For example, there are sex differences in neurocircuitry-level mechanisms of oxytocin control of social and anxiety-related behaviours in rats [93]. Recently, CBD was found to upregulate mRNA of the oxytocin neuropeptide, which may have contributed to its antipsychotic-like effects [94]. Interestingly, the authors proposed that CBD-induced upregulation of oxytocin may exert antipsychotic-like effects and improve social deficits in an animal model of autism-spectrum disorder by modulating endocannabinoid tone [94], highlighting once again the convergence of multiple potential mechanisms of CBD’s neuropsychiatric effects. Another example is noradrenergic control of arousal, which is a particularly salient component of animal models of PTSD. It appears that in female rats, locus coeruleus neurons are more sensitive to activation by corticotrophin-releasing hormone (CRF), leading to increased release of noradrenaline and thus increased arousal, which parallels findings from human studies that women with PTSD and major depressive disorder (MDD) tend to have greater symptoms of hyperarousal [93]. At least two studies have found that CBD can modulate noradrenaline levels, at least in rodent models, and that this may be a mechanism that contributes to the antidepressant-like effects of CBD [95,96]. Thus, sex differences in oxytocin and noradrenaline neurocircuitry may underlie some of the observed sex differences in mood, anxiety, and trauma-related effects of CBD, though much more work is needed to better understand these complex, multifactorial relationships.

In addition to potential neurocircuitry and neurochemical mechanisms of sex differences in the neuropsychiatric effects of CBD, it should also be noted that observed behavioural differences may simply be due to sex differences in CBD pharmacokinetics (discussed in the previous section). Similar to how Tseng and colleagues found that attenuating the hydroxylation of THC to its 11-OH-THC metabolite abolished sex differences in certain behavioural effects of THC [84], it is possible that sex differences in metabolism of CBD could lead to greater concentrations of psychoactive CBD metabolites in females, which in turn could lead to greater behavioural sex differences in certain animal models. As the literature concerning sex differences in neuropsychiatric effects of CBD evolves, it will be especially pertinent to consider CBD pharmacokinetics. 

### 4.5. Could Gender Impact the Clinical Utility of CBD for the Treatment of Neuropsychiatric Disorders?

While most of the focus in psychiatry and neuroscience has been characterizing the role of sex as a biological variable in the neuropsychopharmacology of psychoactive drugs, the sociocultural construct of gender likely has an important role in determining clinical utility. For example, concerns about loss of custody of children may be a particularly salient gendered barrier to medication utilization and adherence in pregnant people with opioid use disorder [97] and other substance use disorders [98]. Dominant forms of masculinity are often associated with traits such as stubbornness, self-reliance, and victim-blaming that can lead to under-reporting of psychiatric disorders (especially mood and anxiety disorders) and lack of treatment seeking [99]. Since there are known gender influences on cannabis use patterns [100] and sex/gender differences in patterns of CBD use specifically [42,43,44], future human studies should consider whether specific gendered dimensions may impact willingness to use CBD and treatment adherence. 

### 4.6. Limitations and Future Directions

There are a few limitations of this scoping review that should be noted. First, we excluded studies not directly relevant to common psychiatric disorders, which meant excluding data from studies of neurological conditions such as epilepsy (the only condition for which CBD is approved and indicated). However, repeating our search terms and replacing the outcome terms with “epilepsy or seizures” retrieved only 11 search results on PubMed (search conducted 22 August 2022), none of which included a sex-based analysis. Replacing the outcome search terms with “pain” yielded 46 results, at least a few of which did seem to contain a sex-based analysis. Given the upward trend in inclusion of sex as a biological variable in animal and human research, largely driven by mandates from funding agencies [47], there will likely be a much greater number of CBD animal studies and human trials including a sex difference analysis; thus, future scoping or systematic reviews with a focus on individual disease endpoints are warranted. For example, it may be worth conducting a review of sex differences in potential analgesic effects of CBD given the number of search results retrieved.

Second, we excluded studies that did not conduct a sex-based analysis. This unfortunately meant excluding several potentially relevant studies that did not include sex as a factor in their analysis, but instead disaggregated data by sex and conducted within-sex comparisons only. We would posit that this is more a limitation of the available literature than of our review. As others have argued [48], testing for within-sex effects of a drug treatment and then qualitatively comparing the effects across sex is not a statistically meaningful way to test for sex differences in drug effects. Unfortunately, Garcia-Sifuentes and Maney (2021) have recently reported that this is a common phenomenon across life sciences disciplines, finding that of 53 included articles that had a factorial design and reported a sex difference, only 16 (30%) actually provided statistical evidence of a sex difference [48]. It is worth noting that single-sex studies are important for understanding the influence of sex on drug effects and treatment outcomes. Ultimately, if there is a sex difference in a physiological outcome, it is due to either direct impact of sex chromosomes (e.g., an outcome influenced by a Y-chromosome gene) or gonadal hormones [88]. Single-sex studies allow for investigation of the mechanism of sex difference, for example, the impact of mouse estrous cycle (and by inference, ovarian hormones) on the effects of CBD on fear memory in female mice [77]. As the literature evolves, it will be imperative to consider not just whether a sex difference exists in a certain effect of CBD, but what the mechanism is. 

Finally, due to the breadth of the literature covered, we did not pre-specify our endpoints of interest when screening articles and extracting data. This makes the presentation of our results less systematic and more subject to bias. However, given the limited number of articles and wide range of outcomes captured by our search protocol, it would not have been feasible to pre-specify outcomes without losing additional endpoint data that were informative to our review. We hope that our scoping review will serve as a starting point to encourage more systematic investigations of the sex and CBD literature as it evolves. 

## 5. Conclusions

While there has been growing interest in evaluating the potential of CBD to treat psychiatric disorders, we report here that there has been little consideration of sex differences in the effects or pharmacokinetics of CBD. Even when sex is considered in factorial designs of CBD effects, it is not often included as a factor, and thus there is no statistical evidence to support the conclusion that the effects of CBD may be “sex-dependent”. Based on the limited available evidence, we conclude that there may be a role of sex in determining long-term consequences of gestational exposure to CBD, which may be related to sex differences in embryonic brain concentrations of CBD (higher in females). In addition, the antidepressant-like effects of CBD may be male-specific (not present in females), though this has only been demonstrated in rodent models. One finding that seemed consistent across preclinical and clinical evidence was increased CBD concentrations in females compared to males, though in humans this was only true in healthy volunteers, not in patient samples. Studies that were excluded from our review have found that sex may impact the effects of CBD on a few relevant outcomes, including neurobiological mechanisms of psychosis and binge drinking in preclinical models. In order to advance understanding of sex differences in CBD effects, more work is needed to encourage optimal factorial designs that can test for an interaction of sex with CBD treatment effect. In addition, as we uncovered very little human data, studies characterizing sex differences in CBD effects in clinical trials are imperative. Finally, in order to best understand the potential clinical utility of CBD in psychiatry, human studies are needed to investigate how gender influences CBD use. 

## Figures and Tables

**Figure 1 biomolecules-12-01462-f001:**
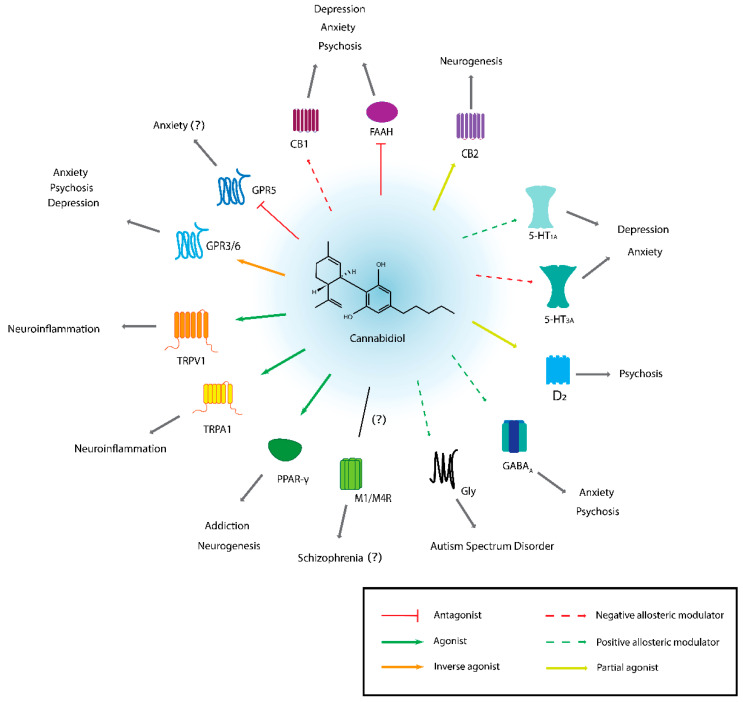
Summary of molecular targets of CBD relevant to psychiatry. For each molecular target, the activity of CBD is indicated by either solid or dotted coloured lines (with the except of the muscarinic receptors M1 and M4, where CBD’s pharmacological activity is not clear). Question marks indicate limited or uncertain evidence. TRPV1 and TRPA1 are both involved in neuroinflammation (which is especially relevant for epilepsy and pain), and this may link their activity to depression, addiction, or other mental health conditions [18], though the evidence is limited. For detailed reviews of CBD molecular targets, the interested reader is directed to references [18,20,26].

**Figure 2 biomolecules-12-01462-f002:**
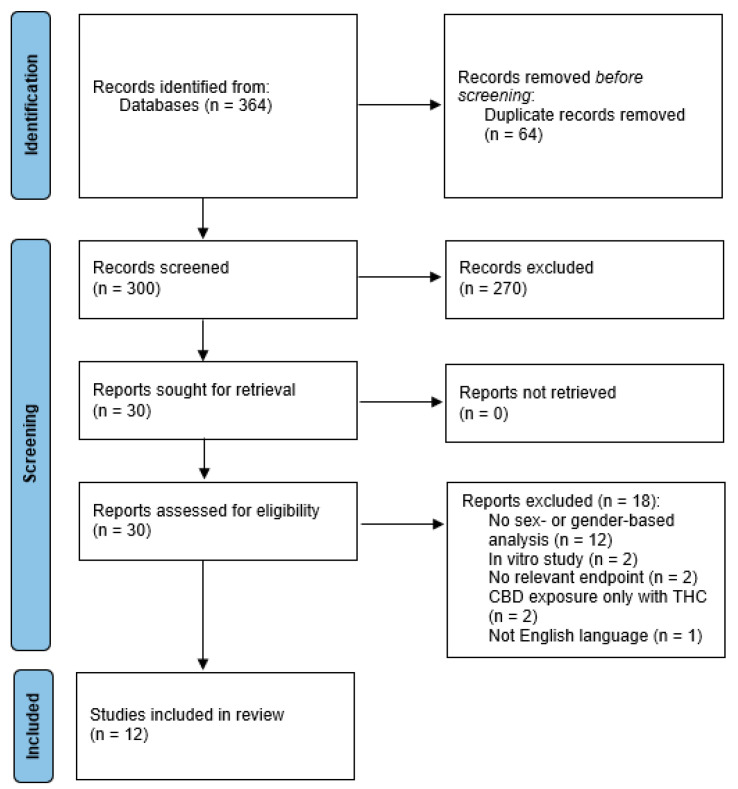
Flow diagram depicting article screening process, presented according to the Preferred Reporting Items for Systematic reviews and Meta-Analyses (PRISMA) [49].

**Table 2 biomolecules-12-01462-t002:** Overview of included articles detailing studies in humans.

Reference	Study Population and Sex (*n*)	CBD Dosing	Relevant Endpoints	Results
Bolsoni et al., 2022 [58] ^1^	Patients diagnosed with PTSD Female *n* = 25 Male *n* = 8	Single oral dose of 300 mg CBD or placebo (parallel design)	Subjective and physiological responses to recalling trauma	No effect of sex
Consroe et al., 1991 [59]	Patients diagnosed with Huntington’s disease Female *n* = 6 Male *n* = 8	6 weeks of 10 mg/kg CBD or placebo daily (crossover design)	Plasma CBD concentrations: (1) average over 6 weeks of treatment; (2) during 1-week washout	No significant sex difference in either measure (1,2)
Contin et al., 2021 [60]	Patients diagnosed with Dravet syndrome or Lennox-Gastaut syndrome Female *n* = 24 Male *n* = 19	1–12 months of treatment with 4.6–22.8 mg/kg daily of CBD (open, prospective design)	Ratio of morning trough CBD plasma concentration to weight-adjusted daily dose	No significant sex difference
Knaub et al., 2019 [61]	Healthy volunteers Female *n* = 8 Male *n* = 8	Single oral dose of 25 mg CBD in one of two formulations, MCT or SEDDS (crossover design)	PK parameters for MCT-CBD: (1) AUC_0–8h_, (2) AUC_0–24h_, (3) C_max_, (4) T_max_; PK parameters for SEDDS-CBD: (5) AUC_0–8h,_ (6) AUC_0–24h_, (7) C_max_, (8) T_max_	MCT-CBD AUCs (1,2) were significantly higher in females SEDDS-CBD T_max_ (8) was significantly faster in males No other sex differences (3–7)

AUC, area under the curve; C_max_, maximum concentration; MCT, medium-chain triglycerides; PK, pharmacokinetic; PTSD, post-traumatic stress disorder; SEDDS, self-emulsifying drug delivery system; T_max_, time to maximum concentration. ^1^ Sex difference analysis was not clearly described in the methods or results; it was presented as an exploratory analysis in the discussion (the data were not shown in the manuscript).

## Data Availability

Not applicable.

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
