# Peer review of "Sex Differences in the Neuropsychiatric Effects and Pharmacokinetics of Cannabidiol: A Scoping Review"

_biomolecules, 2022, doi:10.3390/biom12101462_

Round 1
Reviewer 1 Report
In this work Matheson and colleagues reviewed the influence of sex on CBD neuropsychiatric effects and pharmacokinetics. The manuscript was limited by the reduced studies on the topic, but the authors provided a holistic and focused review with reasonable critical evaluation of the results obtained. I believe this manuscript can be published as it is, but I suggest a minor improvement the authors can use to improve the text: I suggest the authors to better explain the endocannabinoid system for a better understanding of it.
Author Response
We thank the reviewer for their kind appraisal of our manuscript. We have provided more detail about the endocannabinoid system in the introduction to better orient the reader to the ECS (see p1-2, lines 46-58). Note that we have made a few minor revisions throughout the rest of the manuscript to avoid repetition (e.g., revised sentence on p2, lines 87-91 to avoid repetition of FAAH, AEA, and CB2R definitions).
Reviewer 2 Report
In their Review, Matheson et al. gather and synthesize all the evidence available as to the sex-specific effects of CBD in different neuropsychiatric conditions and pharmacokinetic studies. The review is exquisitely written, and the results here summarized are clearly exposed. Despite the breath of original papers, the Review serves its purpose to the reader in collecting all the available evidence in an easily readable manuscript. In spite of my initial enthusiasm, I was left a bit cold after reading the Discussion. I think that the text could have been much more informative if the rationale for expected sex-differences would have considered known sexually-dimorphic brain systems. More below:
Major revision:
· In my opinion, the Discussion feels like a re-write of the Results section but with the addition of conclusions such as “more studies are needed”. I would suggest the Authors to re-think the Discussion to include insights about what neuropsychiatric effects of CBD are expected to present sex differences considering the possible sexual dimorphisms characterizing the systems it targets, as shown in Figure 1. With this, the Discussion will also identify outstanding research questions that may motivate future original works on that direction.
· The Authors should dig deeper into the putative mechanisms that could be explaining the sex-differences already reported.
o For instance, in lines 475-476 a sex-dependent mechanism altering the uptake of CBD is hinted, but no other information is given about what could explain this difference.
o A second example follows in the next paragraph, what are the putative mechanisms that could explain the female-specific effect of adolescent CBD on spatial working memory? An enzymatic sex difference is mentioned, but no explanation is given.
o Also, what sexual dimorphic mechanism/substrate could explain the male-specific anti-depressant effects of CBD?
Author Response
In their Review, Matheson et al. gather and synthesize all the evidence available as to the sex-specific effects of CBD in different neuropsychiatric conditions and pharmacokinetic studies. The review is exquisitely written, and the results here summarized are clearly exposed. Despite the breath of original papers, the Review serves its purpose to the reader in collecting all the available evidence in an easily readable manuscript. In spite of my initial enthusiasm, I was left a bit cold after reading the Discussion. I think that the text could have been much more informative if the rationale for expected sex-differences would have considered known sexually-dimorphic brain systems. More below:
We thank the reviewer for their kind words about our manuscript.
Major revision:
- In my opinion, the Discussion feels like a re-write of the Results section but with the addition of conclusions such as “more studies are needed”. I would suggest the Authors to re-think the Discussion to include insights about what neuropsychiatric effects of CBD are expected to present sex differences considering the possible sexual dimorphisms characterizing the systems it targets, as shown in Figure 1. With this, the Discussion will also identify outstanding research questions that may motivate future original works on that direction.
We thank the reviewer for this comment; we agree, and feel that in responding to this comment, we have substantially improved the manuscript. We have included an additional section (4.4. “Potential Mechanisms of Sex Differences in CBD Effects and Pharmacokinetics”, p16-17) to address this comment. This new section is certainly not meant to be a comprehensive overview of every possible mechanism of sex differences in CBD effects, but gives the reader some indication of where it might be most salient to start to look for mechanisms, based on the very limited available evidence and our own speculation.
- The Authors should dig deeper into the putative mechanisms that could be explaining the sex-differences already reported.
o For instance, in lines 475-476 a sex-dependent mechanism altering the uptake of CBD is hinted, but no other information is given about what could explain this difference.
o A second example follows in the next paragraph, what are the putative mechanisms that could explain the female-specific effect of adolescent CBD on spatial working memory? An enzymatic sex difference is mentioned, but no explanation is given.
In addition to the new section added to speculate on potential mechanisms of sex differences in CBD effects, we have also provided some more discussion of the potential mechanism underlying this particular sex difference (spatial working memory following gestational/developmental exposure to CBD) at the end of the last paragraph of section 4.1 (p14, lines 504-510).
o Also, what sexual dimorphic mechanism/substrate could explain the male-specific anti-depressant effects of CBD?
We have speculated on potential mechanisms in the new section added (4.4; specifically, see p17, lines 641-646 and lines 664-670).
Round 2
Reviewer 2 Report
The Authors have addressed all my comments and concerns.